# Full-Length Transcriptome Profiling of the Complete Mitochondrial Genome of *Sericothrips houjii* (Thysanoptera: Thripidae: Sericothripinae) Featuring Extensive Gene Rearrangement and Duplicated Control Regions

**DOI:** 10.3390/insects15090700

**Published:** 2024-09-14

**Authors:** Qiaoqiao Liu, Shiwen Xu, Jia He, Wanzhi Cai, Xingmin Wang, Fan Song

**Affiliations:** 1Department of Entomology, College of Plant Protection, South China Agricultural University, Guangzhou 510640, China; liuqq@scau.edu.cn (Q.L.); wangxmcn@scau.edu.cn (X.W.); 2MOA Key Lab of Pest Monitoring and Green Management, Department of Entomology, College of Plant Protection, China Agricultural University, Beijing 100193, China; xusw2019@126.com (S.X.); hejiayc@126.com (J.H.); caiwz@cau.edu.cn (W.C.); 3Ningxia Key Lab of Plant Disease and Pest Control, Institute of Plant Protection, Ningxia Academy of Agriculture and Forestry Science, Yinchuan 750002, China

**Keywords:** gene rearrangement, full-length transcriptome, mitochondrial gene transcription, RNA processing

## Abstract

**Simple Summary:**

A total of 37 mitochondrial genes appears in a certain arrangement in most insects, and the transcription of this conserved mitochondrial genome (mitogenome) has been well-studied in species like *Drosophila melanogaster*, *Erthesina fullo* and *Coridius chinensis*. However, transcription of the mitogenome with extensive gene rearrangement has rarely been studied. In this research, we sequenced the mitogenome and mitochondrial transcriptome of *Sericothrips houjii* (Thysanoptera: Thripidae: Sericothripinae). The mitogenome of *S. houjii* exhibited extensive gene rearrangement and contained two control regions (CRs) with sequence repeats. Compared to the insect mitogenome with a typical gene order, the mitogenomic feature, relative gene expression level, transcriptional model and post-transcriptional cleavage of *S. houjii* were quite different. Unlike other insects where ribosomal RNA (rRNA) is typically highly expressed, *ND4/ND4L* in *S. houjii* exhibits the highest expression. Both strands of this mitogenome were entirely transcribed and the bicistronic messenger RNA (mRNA) *COI/ND3* was reported for the first time in insects. Our study provides new insights into the transcriptional and post-transcriptional regulation processes in the insect mitogenome with extensive gene rearrangement and duplicated CRs.

**Abstract:**

The mitochondrial genome (mitogenome) of Thysanoptera has extensive gene rearrangement, and some species have repeatable control regions. To investigate the characteristics of the gene expression, transcription and post-transcriptional processes in such extensively gene-rearranged mitogenomes, we sequenced the mitogenome and mitochondrial transcriptome of *Sericothrips houjii* to analyze. The mitogenome was 14,965 bp in length and included two CRs contains 140 bp repeats between *COIII-trnN* (CR1) and *trnT-trnP* (CR2). Unlike the putative ancestral arrangement of insects, *S. houjii* exhibited only six conserved gene blocks encompassing 14 genes (*trnL2-COII*, *trnD-trnK*, *ND2-trnW*, *ATP8-ATP6*, *ND5-trnH-ND4-ND4L* and *trnV-lrRNA*). A quantitative transcription map showed the gene with the highest relative expression in the mitogenome was *ND4-ND4L*. Based on analyses of polycistronic transcripts, non-coding RNAs (ncRNAs) and antisense transcripts, we proposed a transcriptional model of this mitogenome. Both CRs contained the transcription initiation sites (TISs) and transcription termination sites (TTSs) of both strands, and an additional TIS for the majority strand (J-strand) was found within antisense *lrRNA*. The post-transcriptional cleavage processes followed the “tRNA punctuation” model. After the cleavage of transfer RNAs (tRNAs), *COI* and *ND3* matured as bicistronic mRNA *COI/ND3* due to the translocation of intervening tRNAs, and the 3′ untranslated region (UTR) remained in the mRNAs for *COII, COIII, CYTB* and *ND5*. Additionally, isoform RNAs of *ND2*, *srRNA* and *lrRNA* were identified. In summary, the relative mitochondrial gene expression levels, transcriptional model and post-transcriptional cleavage process of *S. houjii* are notably different from those insects with typical mitochondrial gene arrangements. In addition, the phylogenetic tree of Thripidae including *S. houjii* was reconstructed. Our study provides insights into the phylogenetic status of Sericothripinae and the transcriptional and post-transcriptional regulation processes of extensively gene-rearranged insect mitogenomes.

## 1. Introduction

The insect mitochondrial genome (mitogenome) is a double-strand circular DNA, which typically includes 13 protein-coding genes (PCGs), 22 transfer RNA genes (tRNAs), 2 ribosomal RNA genes (rRNAs) and a control region (CR) that plays a key role in regulating replication and transcription [1,2]. In many insect groups, including those with earlier origins, these components are arranged in a compact manner with a typical order, which is inferred as the ancestral gene arrangement of insect mitogenomes (hereinafter referred to as “ancestral arrangement”) [3,4,5]. With the rapid advancement of DNA sequencing technology, the duplication of CRs and gene rearrangements in various insect taxa have been increasingly characterized as more insect mitogenomes are sequenced [6,7,8]. To date, studies on insect mitogenomes have primarily focused on the aspects such as base composition, codon usage, gene arrangement and phylogenetic evolution [4,9,10], while studies on their transcriptional model and post-transcriptional processing have been limited to few species. Five transcriptional cassettes in *Drosophila* have been inferred by Berthier et al. [11] and confirmed by Stewart et al. through 5′ and 3′ RACE (rapid amplification of cDNA ends), circularization and RT-PCR (reverse transcription polymerase chain reaction) methods [12]. Roberti et al. [13] discovered the *Drosophila* mitochondrial transcription termination factor (DmTTF) binding to two transcription termination sites (TTSs) and proposed two models of mitogenomic transcription. The difference among these models was whether the gene blocks of *astrnS2-asCYTB-asND6* and *astrnF-asND5-astrnH-asND4-asND4L* have been transcribed. The post-transcriptional cleavage process was thought to follow the reverse cleavage of the “tRNA punctuation” model, which suggested tRNAs were cleaved from 3′ end to 5′ end in the primary transcript to release the messenger RNA (mRNA) and rRNA [12,14].

Recently, the full-length transcriptome has been utilized to profile mitochondrial gene expression in insects. In *Erthesina fullo*, five primary transcripts were mapped into the five transcriptional cassettes of *Drosophila*, two novel long non-coding RNAs (lncRNAs) of CR have been detected and both the reverse cleavage and forward cleavage model (tRNAs were cleaved from 5′ end to 3′ end) have been reported [15,16]. Further studies investigated the mitochondrial transcriptional models in *Coridius chinensis* and *Aphidius gifuensis* suggest that the mitogenome is transcribed continuously from CR, producing an almost complete primary polycistronic transcript from both strands [17,18]. The only exception is that the antisense genes downstream of *trnS2* are not transcribed. Each strand has been transcribed into one primary polycistronic transcript except the majority strand (J-strand) in the *A. gifuensis* mitogenome, which forms at least five primary polycistronic transcripts as there were five TTSs recognized. In *A. gifuensis*, the gene clusters of CR-*trnI-trnM-trnQ* and *trnW-trnC-trnY* have been rearranged as *trnI*-CR1-*trnM*-CR2-*trnQ* and *trnW-trnY-trnC*, but there is no repeated sequence between the two CRs [18]. These rearranged gene clusters seem to have little effect on the multiple TTSs of the J-strand, and the TISs of two strands were distributed in different CRs. It remains unclear whether more extensive gene rearrangement and duplicated CRs affects transcription, post-transcriptional cleavage and the number of TIS and TTS.

Compare to the ancestral arrangement, the thysanopteran mitogenome is well-known for its extensive gene rearrangement, with only 10–18 relative conserved genes encompassed by 3–6 gene blocks [19]. Therefore, the mitogenome of Thysanoptera serves as an excellent model for studying the processes of mitogenomic transcription in an extensive gene rearrangement. In this study, we sequenced the mitogenome and the full-length mitochondrial transcriptome of *Sericothrips houjii*. Firstly, we precisely annotated the mitogenome of *S. houjii* by using the information of mitochondrial transcripts and illustrate the characteristics of mitogenome. Next, we constructed a quantitative transcription map using high-quality full-length mitochondrial transcripts which were generated from single-molecule, real-time (SMRT) sequencing. Then, the polycistronic RNAs, mature RNAs and isoform RNAs were counted in detail. Finally, the models of mitochondrial transcription and post-transcriptional cleavage in *S. houjii* were proposed and compared with other insects. These results revealed the unique features of mitochondrial transcripts in *S. houjii* and offered valuable insight into the patterns of gene transcription and post-transcriptional regulation in a mitogenome with extensive gene rearrangement.

Currently, four subfamilies, Panchaetothripinae, Dendrothripinae, Sericothripinae and Thripinae, are recognized in Thripidae [20]. The phylogenetic analyses based on morphology and several molecular loci support Panchaetothripinae, Dendrothripinae and Sericothripinae nested within Thripinae [21,22]. Meanwhile, the phylogenetic analyses based on mitochondrial genes support only Sericothripinae nested within Thripinae [19,23,24], indicating that the mitochondrial genes are useful to recover the monophyly of the subfamilies. However, only one mitogenome of a species, *Neohydatothrips samayunkur*, from Sericothripinae has been published to date. The monophyly and phylogenetic status of Sericothripinae remains unclear. Here, we sequenced the mitogenome from *Sericothrips*, which is the type genus of Sericothripinae. To further understand the phylogenetic status of Sericothripinae, the phylogeny of Thripidae was analyzed using Aeolothripidae as the outgroup based on the published mitogenomes. Our study provides insights into the taxonomic decision and phylogeny of Thripidae.

## 2. Materials and Methods

### 2.1. Sample Collection

Specimens of *S. houjii* were collected from alfalfa (*Medicago sativa*) in Hongsibao, Wuzhong, Ningxia, China, in August 2020. Living thrips were frozen by drikold (−80 °C) for transcriptome extraction or soaked in absolute ethyl alcohol stored at −20 °C for total genomic DNA extraction. Voucher specimens were preserved at the Entomological Museum of China Agricultural University, Beijing, China.

### 2.2. Acquisition and Annotation of Mitogenome

The total genomic DNA were extracted from seven adults using the DNeasy Blood and Tissue kit (Qiagen, Dusseldorf, Germany) according to the manufacturer’s protocol. An Illumina TruSeq library was prepared with an average insert size of 350 bp and sequenced on the Illumina Hiseq 6000 platform (Berry Genomics, Beijing, China) with 150 bp paired-end reads. A total of 6 Gb raw data was obtained and adapters were trimmed from raw reads using Trimmomatic [25]. The short and low-quality reads were removed by using Prinseq v0.20.4 [26] with the parameter poly-Ns > 15 bp, <75 bp in length and quality score < 3. The remaining clean data were used to *de novo* assemble by IDBA-UD [27], with overlapping similarity > 98%, and minimum and maximum *k* values of 45 bp and 145 bp, respectively. A 541 bp fragment of the *COI* gene (GenBank accession number is HQ605969) was used to identify the mitochondrial scaffold using BLASTN [28] searches with at least 98% similarity. To confirm the accuracy of the assembly, clean reads were mapped to the obtained mitogenome using Geneious v10.1.3 (http://www.geneious.com/, accessed on 15 September 2019), with mismatches of up to 2%, a maximum gap size of 3 bp and a minimum overlap of 40 bp. Finally, the complete mitogenome of *S. houjii* was obtained with a 35,545× average sequencing depth.

The mitogenome of *S. houjii* was preliminarily annotated by MitoZ [29] with settings “genetic_code 5” and “clade Arthropoda”, and further corrected by alignment with homologous mitochondrial genes of *Neohydatothrips samayunkur* (Thysanoptera: Thripidae: Sericothripinae. GenBank accession number is MF991901).

### 2.3. Acquisition of Mitochondrial Transcriptome

A total of 50 adults of *S. houjii* were pooled for total RNA extraction, using TRIzol Universal (Tiangen, Beijing, China). The concentration and purity of total RNA was detected by NanoDrop 2000 spectrophotometry (Thermo Fisher, Waltham, MA, USA), and the integrity was detected by the Agilent 2100 system (Agilent Technologies, Santa Clara, CA, USA). Total RNA was reverse transcribed into cDNA using the Clontech SMARTer PCR cDNA Synthesis Kit (Clontech Laboratories, Inc., Mountain View, CA, USA) with 3′ SMART CDS Primer II A (5′-AAGCAGTGGTATCAACGCAGAGTAC-T_(30)_-3′) and 5′ SMARTer II A Oligonucleotide (5′-AAGCAGTGGTATCAACGCAGAGTACATGGG-3′). The cDNA was amplified and constructed a library with an insert size between 1 and 10 kb was constructed according to the PacBio IsoSeq protocol after size selection. Finally, one SMRT cell was performed on the PacBio Sequel platform with circular consensus sequencing (CCS) mode at Berry Genomics Company (Beijing, China).

The PacBio SMART Analysis v10.2 (http://www.pacb.com/devnet/, accessed on 17 December 2021) was performed on the raw data. Of which, CCS v5.0.0 with parameters (Minimum Full Passes = 1, Minimum Predicted Accuracy = 0.9) was used to process the sequenced reads into the high-quality circular consensus sequencing (CCS) reads, lima v2.0.0 was used to produce full-length non-chimera (FLNC) reads also known as draft transcripts by removing the full-length non-chimera, primers of cDNA and polyA sequences in 3′ end, and Samtools v1.11 (http://www.htslib.org/, accessed on 18 December 2021) was used to read the sequences in the binary file (BAM file).

### 2.4. Mitochondrial Transcript Identification

The mitochondrial transcripts were mapped to the circular topology mitogenome using Geneious v10.1.3, with maximum gap per read = 10%, maximum gap size = 8, minimum overlap identity = 90%, maximum mismatches per read = 5%, and maximum ambiguity = 5. The coverage depth of each base in the reference mitogenome were calculated to the coverage histogram, namely, quantitative transcription map. The 5′ and 3′ ends of mature transcripts, polycistronic transcripts and antisense transcripts were identified and classified after precise modification of the mitogenome according to the full-length transcripts. The representative transcripts from each of the same type were plotted on the quantitative transcription map. The mature and precursor transcripts were identified by aligning the boundaries of genes or gene blocks. The transcripts with 5′ and/or 3′ ends truncated (not mapped to the gene boundaries of the gene or gene block) were considered as intermediate degradation products. Isoform RNA can be identified by multiple transcripts of a single gene with identical 5′ and 3′ ends, but the 5′ and/or 3′ ends are not aligned to the gene boundary.

### 2.5. Phylogenetic Analysis

The amino acid of each PCG and nucleotide of each rRNA were individually aligned using MAFFT [30]. The poorly aligned sites of rRNA were removed using trimAL [31] with the parameter “-automated1”, while additional parameters “-ignorestopcodon -backtrans” was used to remove the stop codon and back translation of the amino acid sequence. The PCG alignments excluded the third codon position in MEGA 7.0.21 [32]. Alignment concatenation was performed in Geneious v10.1.3 (http://www.geneious.com/, accessed on 15 September 2019) to generate the datasets PCG_rRNA (13 PCGs and two rRNAs) and PCG12_rRNA (13 PCGs with the third codon position excluded and two rRNAs) for phylogenetic analyses. A site-heterogeneous mixture model (CAT + GTR) implemented in PhyloBayes avoids the false grouping of unrelated taxa with similar base composition and accelerated evolutionary rate in the reconstruction of the phylogeny in Thysanoptera [19]. Thus, the datasets were analyzed under a CAT+GTR model using Phylobayes MPI 1.8c [33] on the CIPRES Science Gateway [34]. In each analysis, two independent chains starting from a random tree were run for 30,000 cycles, with trees being sampled every cycle until 30,000 trees were sampled. The initial 7500 trees of each MCMC run were discarded as burn-in. A consensus tree was generated from the remaining trees combined from two runs.

## 3. Results

### 3.1. The Mitogenome Feature of Sericothrips houjii

We annotated the mitogenome of *S. houjii* using MitoZ and homologous alignment, and further modified the annotations precisely according to the full-length transcripts. The annotation with MitoZ and homologous alignment are based on the nucleotide alignment, the mitochondrial gene code of PCG and the cloverleaf structure of tRNA [29]. MitoZ was precise in the gene arrangement and homologous alignment corrected the translation frame, but did not predict the gene boundaries well. Compared to the annotation results of MitoZ and homologous alignment, the boundary of seven PCGs (*ND3*, *COIII*, *CYTB*, *ND2, ND1, ATP6* and *ND6*), two tRNAs (*trnR* and *trnT*) and two rRNAs (*sRNA* and *lrRNA*) was modified according to the full-length transcripts (Table 1). After modification, the overlapping regions between *ND3* and *trnL2*, *trnF* and *srRNA*, *srRNA* and *ATP8*, *trnV* and *lrRNA*, and *lrRNA* and *trnS1* were eliminated, and the intergenic spacers between *COII* and *trnR*, *trnK* and *COIII*, *trnI* and *CYTB*, *trnW* and *ND1*, *ND2* and *trnW*, *trnL2* and *trnT* and *trnC* and *ND6* were reduced or disappeared, which recovered the compactness of this mitogenome.

The mitogenome of *S. houjii* is a single circular molecule 14,965 bp in length (GenBank accession number is PP697967), which contains 37 encoding genes (13 PCGs, 22 tRNAs and 2 rRNAs) and two putative control regions (CR1 and CR2). The minority strand (N-strand) encodes 3 protein-coding genes (*ND4*, *ND4L* and *ND5*) and three tRNAs (*trnY*, *trnP* and *trnH*), whereas the J-strand encodes the remaining 31 genes. This mitogenome contains a repeat region of 230 bp in length, which covered 144 bp of CR1 and 220 bp of CR2. The 76 bp repeats before CR1 are a part of the complete *COIII* genes but were joined to CR2 as a pseudogene (Figure 1). The motifs of the control region located at the repeat region include G(A)nT, polyT/polyA, A(T)n, stem-loop and (TA)n.

Compared with the ancestral arrangement, the gene order of the *S. houjii* mitogenome underwent extensive rearrangement (Figure A1). At least ten rearrangement events were proposed in the evolution of the *S. houjii* mitogenome, with eight genes (*trnQ*, *ND1*, *trnF*, *srRNA*, *trnL1*, *trnC*, *trnV* and *lrRNA*) inverted and only six relative conserved gene blocks (*trnL2-COII*, *trnD-trnK*, *ND2-trnW*, *ATP8-ATP6*, *ND5-trnH-ND4-ND4* and *trnV-lrRNA*) remaining (Figure A1). Gene blocks *COI-ND3*, *trnR-trnG*, *COIII-CR1-trnN-trnE-trnQ-trnA-trnS1-trnI-CYTB-trnY*, *ND1-trnM-trnF-srRNA*, *trnL1-trnT* and *trnC-ND6* were newly generated in the processes of mitochondrial gene rearrangement.

### 3.2. The Quantitative Transcription Map of Sericothrips houjii Mitogenome

We obtained a total of 3.32 Gb high-quality CCS (accuracy ≥ 0.9) with 1,144,294 transcripts from the raw data of the full-length transcriptome sequencing, and 1,005,191 draft transcripts totaling 2.18 Gb. In these draft transcripts, 37,040 transcripts (GenBank accession number is PRJNA1130360) were mapped to the complete mitogenome of *S. houjii* with a mean coverage of 4796.7× in depth. The quantitative transcription map showed that the transcripts of the N-strand (19,126) were more abundant than that of the J-strand (17,914) (Figure 2). We found that the relative expression levels of mitochondrial genes, ranking from highest to lowest, were *ND4*, *ND4L*, *ND3*, *COI*, *srRNA*, *COIII*, *lrRNA*, *CYTB*, *ND1*, *ND5*, *ND2*, *ATP6*, *ATP8*, *COII* and *ND6*.

### 3.3. Polycistronic RNA and Precursor RNA Cleavage

In addition to *ATP8/ATP6* and *ND4/ND4L* maturing as bicistronic mRNA like other insects [12,15,17,18], another bicistronic mRNA *COI/ND3* was recognized in the *S. houjii* mitogenome, which was firstly identified in the insect mitogenome. Excluding degraded transcripts, there were 584 sense polycistronic transcripts (except *COI/ND3*, *ATP8/ATP6* and *ND4/ND4L*) mapped to the mitogenome of *S. houjii*, of which six transcripts were transcribed from J-strand can be divided into four types and 578 were transcribed from N-strand can be divided into 25 types (Figure 2). These polycistronic transcripts covered 10 PCGs (*ATP8*, *ATP6*, *COI*, *COII*, *COIII*, *CYTB*, *ND1*, *ND2*, *ND3* and *ND6*), 2 rRNAs (*srRNA* and *lrRNA*) and 17 tRNAs (*trnL2*, *trnR*, *trnD*, *trnK*, *trnN*, *trnE*, *trnQ*, *trnA*, *trnS1*, *trnI*, *trnW*, *trnM*, *trnF*, *trnT*, *trnC, trnV* and *trnS2*). We found the polycistronic transcripts *trnS2/COI/ND3*, *trnL2/COII*, *trnI/CYTB/astrnY/ND2*, *trnW/ND1*, *trnC/ND6* and *trnV/lrRNA* lack their downstream tRNAs in 3′ end, indicating that their primary transcript had the ability to remove tRNAs from 3′ end to 5′ end. However, the polycistronic transcripts *COI/ND3/trnL2/COII*, *trnD/trnK/COIII/CR** (* means the 3′ end or 5′ end of the transcript was not aligned to the gene boundary) and *trnK/COIII/CR** indicating the *trnS2* and *trnD* were removed from 5′ end to 3′ end. By comparing the related mature mRNAs and rRNAs of the polycistronic transcripts, it was found that the cleavage site of polycistronic transcripts was the gene boundaries of tRNAs. After cleave the RNAs from polycistronic transcripts, the untranslated regions (UTRs) transcribed from the non-coding regions (NCRs) at the 5′ ends of *COIII*, *CYTB*, *ND2*, *ND4L/ND4* and *ND6* were removed to expose the start codon. However, the 3′ ends of the mRNAs *COII*, *CYTB* and *ND5* were connected to the 5′ end of their downstream tRNA, wherein the 3′ UTR was retained. Beyond the situation, the 3′ UTRs of *COIII* transcripts were variable in length from 5 to 120 bp, and the polyadenylate sites were optional from nucleotide position 3872 to 3987.

### 3.4. Isoform RNA

We identified in total 153 mature transcripts of *ND2* (ChrM: 5678–6659). Among them, 18 transcripts ended at position 6582 and 135 transcripts ended at position 6659, resulting in two isoforms of *ND2* transcripts. The isoform1 of *ND2* mRNA is 982 bp in length (ChrM: 5678–6659) and isoform2 is 899 bp (ChrM: 5678–6582). Excluding the degraded and polycistronic transcripts, 4023 transcripts were mapped to *srRNA* (ChrM: 7778–8500). The 5′ ends of these transcripts were mapped to position 7778 but the 3′ ends were mapped to 17 different positions (Figure 3). Considering the nucleotide error of PacBio Sequel sequencing and the inconsistency of polyA length at the 3′ end of the transcripts, we classified these transcripts into eight isoform types (isoform1-8) according to the nearby 3′ end. Each isoform type of *srRNA* was supported by multiple transcripts (Figure 3). For *lrRNA* (ChrM: 13,820–14,902), 397 mature transcripts were detected. The 5′ ends of these transcripts were mapped to position 13,820 or 13,942, but the 3′ ends were mapped to positions 14,902, 14,620 and 14,243, which is supported by 22 transcripts (isoform1, ChrM: 13,820–14,243), 62 transcripts (isoform2, ChrM: 13,820–14,620), 116 transcripts (isoform3, ChrM: 13,820–14,902), 64 transcripts (isoform4, ChrM: 13,942–14,620) and 133 transcripts (isoform5, ChrM: 13,942–14,902), respectively. In addition, the 3′ ends of three polycistronic transcripts *trnW/ND1/trnM/trnF/srRNA** (see nos. 145884403, 159975201 and 19530688) were identical with isoform1 and isoform4 of *srRNA* and 32 bicistronic transcripts *trnV/lrRNA** were identical with the isoforms of *lrRNA*. These isoforms are easy to distinguish from the degraded transcripts as multiple transcripts were identical in 5′ and 3′ end.

The 5′ ends of the degraded transcripts from *ND2* were variable within genes, while the 3′ ends were identical in two isoforms, indicating that the *ND2* was degraded from the 5′ end to 3′ end. However, the tricistronic transcripts of *CYTB/astrnY/ND2** indicated the degradation from the 3′ end to 5′ end. In this study, we detected the 5′ ends of *srRNA* transcripts were variable in positions between 7778 (5′ end of *srRNA*) and 8186, and the 3′ ends of *srRNA* transcripts were variable in positions between 8013 (3′ end of isoform1) and 8500 (3′ end of isoform8), which indicates the *srRNA* were degraded from both ends. The *lrRNA* isoforms had two 5′ end positions (13,820 and 13,942) and three 3′ end positions (14,243, 14,620 and 14,902). The trapezoid formed by the *lrRNA* transcripts in the quantitative transcription map indicates a significantly exonucleolytic degradation from 5′ end to 3′ end (Figure 2). A small number of *lrRNA* transcripts mapped their 5′ end to position 13,820 or 13,942 but were variable in the 3′ end, which can be regarded as the exonucleolytic degradation from 3′ end to 5′ end. In addition, polycistronic transcripts of *trnI/CYTB/astrnP*/*ND2**, *CYTB/astrnP*/*ND2** and *trnW/ND1/srRNA** and bicistronic transcripts of *trnV/lrRNA** indicate that the cleavage of polycistronic transcripts and exonucleolytic degradation can proceed simultaneously.

### 3.5. The Proposed Model of Mitochondrial Transcription

The ncRNA transcribed from CR was considered the precursor of the transcription initial RNAs (tiRNAs) [35] and the polyA/polyT motifs in CR are the signal to initiate the transcription of mitogenome [36]. Three ncRNAs mapped to the J-strand of CRs and one ncRNA mapped to the N-strand of CR in *S. houjii* mitogenome contained the motifs, which was considered as regulating the transcription of mitogenome (Figure 1). So, we proposed both CRs contained the TISs of both strands. As the 5′ end of ncRNAs was located within the *COIII* when the 3′ end was mapped within CR1, whether the TIS was function in the transcription of N-strand remains unclear. The relative expressions from *asATP6* to *astrnN* and from *asCOIII* to *astrnC* were significantly lower than their upstream region (from *trnP* to *ND4L*), indicating CRs may weakly function in the transcription initiation but function well in the termination of the J-strand. In addition, the 5′ end of polycistronic transcript *aslrRNA*/astrnV/asND6/astrnC/ND4L/ND4* (see no. 29428453) and three antisense transcripts *aslrRNA*/astrnV/asND6/astrnC* (see nos. 108333364, 108333364 and 156437306) have the same 5′ end position, ChrM-13,864, which may be another TIS of the J-strand.

The 3′ end of 2175 transcripts of *COIII*, and the 3′ end of 10 tricistronic transcripts *trnD/trnK/COIII* and bicistronic transcripts *trnK/COIII* were mapped to the repeat region in CR1 (position 3887 to 3968, and position 3987), indicating CR1 contained a TTS of the N-strand. The 3′ end of the transcripts *trnT/CR2** (see nos. 138282373, 28902972 and 59901598) were mapped to the repeat region in CR2 (position 9601, 9607, 9651 and 9673), indicating CR2 contained a TTS of the N-strand. Therefore, both CRs were functional in the termination of N-strand transcription. Similarly, all antisense transcripts of the gene block *CYTB-trnI-trnS1-trnA-trnQ-trnR-trnN*-CR1 were ended within the repeat region of CR1 and the transcript *ND5*/trnP/*CR2* ended within the repeat region of CR2, indicating both CRs were functional in the termination of J-strand transcription.

Based on the above analyses, we proposed the transcriptional model of the *S. houjii* mitogenome (Figure 4). Both the J-strand and N-strand contain at least two TISs and TTSs in the CRs, respectively. The J-strand has an additional TIS within *aslrRNA* to enhance the transcription of the gene block *ND5-trnH-ND4-ND4L*. The mitogenome of *S. houjii* was transcribed into a complete primary polycistronic transcript as CR1 was fully transcribed from N-strand (see no. 83820671) and CR2 was fully transcribed from both strands (see nos. 70518759 and 114557833). The antisense transcripts of the gene blocks *trnR-trnG-trnD-trnK-COIII* and *srRNA-ATP8-ATP6* were not detected, which may be degraded quickly due to their non-functionality. Further research was needed to determine the presence and exact location of TISs and TTSs, such as 5′ and 3′ RACE, RT-PCR and so on.

### 3.6. Phylogenetic Analysis and Mitochondrial Gene Rearrangement of Sericothripinae

A total of 46 mitogenomes were used for phylogeny analyses, including 37 species from Thripidae and 9 outgroups from Aeolothripidae (Table A1). The two phylogenetic trees based on two datasets (PCG_rRNA and PCG12_rRNA) and inferred from the PhyloBayes analysis showed the congruent relationships. As for the results, *S. houjii* is sister to *N. samayunkur*, and the monophyly of Sericothripinae was supported (Figure 5A). Sericothripinae is nested within Thripinae and sister to the group of (*Echinothrips* + *Scirtothrips*).

Then, we compared the gene arrangement between *S. houjii* and its related species, *N. samayunkur* (Figure 5B). Gene blocks of *trnI-CYTB-trnY-ND2-trnW-ND1-trnM-trnF-srRNA-ATP8-ATP6-trnL1-trnT* and *ND5-trnH-ND4-ND4L* were conserved in two mitogenomes, while the gene blocks of *COI-ND3-trnL2-COII*, *trnD-trnR-trnG-trnK-COIII-trnN-trnE* and *ND6-lrRNA-trnS2* in *N. samayunkur* were divided by tRNA translocation in *S. houjii*. The rearrangement degree between these two mitogenomes was calculated in CREx [37], and the results showed that there were six tRNAs (*trnQ*, *trnA*, *trnV*, *trnC*, *trnL2* and *trnS1*) translocated with a breakpoint distance of 15. Mitochondrial gene arrangement in Sericothripinae is diverse like in other thrips taxa.

## 4. Discussion

Mitochondrial gene rearrangement has been deemed to generate the duplication of genes or CR duplication, change in relative gene order, redundancy of pseudogenes and modification of intergenic spacers [38,39,40]. The mitogenome of Thysanoptera have been known for extensive gene rearrangement, a duplicatable control region and highly variable NCRs [19,41]. Due to the rapid evolution, the insertions and deletions of nucleotides may change the length of genes and intergenic spacers. As a result, transcript, which serve as information carriers of gene expression, can annotate the mitogenome more accurately than gene structure predication and homologous sequence alignment. Compare with the annotation according to MitoZ and homologous alignment, the annotations according to transcripts modified the overlaps and intergenic spacers of insect mitogenomes and recovered their compactness. The relative mitochondrial gene expression level was similar in the mitogenomes of *E. fullo*, *C. chinensis* and *A. gifuensis*. These species were found to have the highest expression level in *lrRNA,* and cytochrome c subunits constantly had higher expression levels than NADH-dehydrogenase [15,17,18]. However, the relative expression level was significantly different in *S. houjii* mitogenome. *ND4*-*ND4L* expression was highest, *COI-ND3* and *COIII* expression was higher than the remaining NADH-dehydrogenase and rRNAs and the expression of *COII* was only higher than *ND6*. Extensive gene rearrangement and duplicated CR changed the relative distance between the TISs and genes, which may affect the relative efficiency of post-transcriptional cleavage that affects the relative expression level of genes. In addition, the duplicated sequences produced during gene rearrangement, such as gene duplication and random lose (TDRL), may remain as NCR between genes [19,42]. The NCRs of the *S. houjii* mitogenome have been transcribed as the UTRs of the transcripts, the 5′ UTRs that were removed may be related to the translation need an initiation codon, while the 3′ UTRs remain in the mature mRNA of *COII*, *COIII*, *CYTB* and *ND5*. It was suggested that the 3′ UTR of mRNA is related to the stability of mRNA by forming a secondary structure with a polyadenylate tail [43,44], which maybe another reason for the relative gene expression level change in the mitogenome with extensive gene rearrangement.

There are two theories regarding to the mechanism of isoform formation. One is exonucleolytic degradation, because the isoforms of the same gene have identical 5′ or 3′ ends, which has been thought to fit the exonucleolytic degradation with two orientations [17]. The other is transcription termination, because the corresponding polycistronic transcript have been detected, and there was no transcript mapped to the downstream of the 3′ end of the isoforms [18]. The rRNA isoforms of *S. houjii* were observed to degrade from both ends, and the endings of degraded transcripts were observed to be variable and located between the endings of the isoforms. Thus, the isoforms of the RNAs in the *S. houjii* mitogenome were more likely generated by exonucleolytic degradation from 3′ end to 5′ end or/and 5′ end eto 3′ end instead of the termination of transcription. Neither the 5′ end nor the 3′ end of the degraded mRNA of *ND2* were located between the 3′ ends of the two isoforms. It seems that the isoform2 of *ND2* was generated by transcription termination. But the degradation from 3′ end to 5′ end was observed in the tricistronic transcript of *CYTB/astrnY/ND2*. Although isoform2 of *ND2* was probably generated by exonucleolytic degradation from 3′ end to 5′ end, but the exonucleolytic degradation occurred before the cleavage of *CYTB/astrnY/ND2* and the 3′ ends of two isoforms are so close in distance that the downstream transcript is too short to sequence or degrade quickly.

The functionality of the truncated stop codon T or TA of PCGs are probably recovered by the post-transcriptional polyadenylation [45]. Although the 3′ end of the transcripts polyadenylated, the stop codon in isoform2 of *ND2* is missing, indicating that the isoform2 of *ND2* may not be translated accurately. The evolution rate of the mitochondrial gene is much higher than that of nuclear gene [46] and the mitogenome with extensive gene rearrangement shows a higher rate of nonsynonymous substitution [6,19], but the mitochondrial genes were still homologous and have the relatively conserved domains. Isoform was ubiquitous in the rRNA transcripts of the insect mitochondrial transcriptome, but the isoform type and the orientation of exonucleolytic degradation were different, and the 3′ ends of isoforms were mapped to the nonhomologous regions of genes and the length of the truncated RNAs was variable [15,17,18]. Therefore, the truncated RNAs may not be able to perform normal physiological functions.

Compared with the transcriptional models of the insect mitogenome represented by *Drosophila*, in which the both strands were almost entirely transcribed except the NCR downstream *trnS2* on the N-strand [13,15,18], both strands of the *S. houjii* mitogenome were entirely transcribed, even the control regions. Both CRs were observed to have the initiation potential of mitogenome transcription, but the CR1 was doubtful as its initiative function needs to combine a part sequence of *COIII*. Similarly to the *Drosophila* mitochondrial genome, in which the TIS located at the *ND6* of the *Drosophila* increases the expression of gene block *trnP-trnT-ND4L-ND4-trnH-ND5-trnF* [11,13], the J-strand TIS located within *aslrRNA* (position 13,864) of *S. houjii* increased the expression of the gene block *ND4L-ND4-trnH-ND5-trnP*. This indicates that the J-strand TIS is located upstream of the major template region, regardless of whether the flanking genes were rearranged. Although the *A. gifuensis* mitogenome possessed two CRs, both strands were proposed to have only one TIS. However, both strands of the *S. houjii* mitogenome were proposed to have at least two TISs. The TIS of the mitogenome seems primarily influenced by the duplication of the control region and the location of a major template region. Both CRs of *S. houjii* mitogenome contained the TTS of the J-strand and N-strand, suggesting that the CRs contain the mitochondrial transcription termination factor (mTTF) binding sites. But the mTTF bound to CRs would only serve as an attenuator of the transcription instead of a strict terminator because both CRs were transcribed entirely.

The mitochondrial post-transcriptional cleavage process followed the “tRNA punctuation” model [14,45] in *S. houjii*, which was similar to the patterns observed in other insects [12,15,17,18]. The cleavage processes of six transcripts, *trnS2/COI*, *trnL2/COII*, *trnI/CYTB/astrnY/ND2*, *trnW/ND1*, *trnC/ND6* and *trnV/lrRNA*, followed the “reverse cleavage” model proposed in *Drosophila* [12]. While the cleavage processes of the transcripts *COI/ND3/trnL2/COII*, *trnD/trnK/COIII/CR** and *trnK/COIII/CR** followed the “forward cleavage” model, which was raised in *E. fullo* [15]. However, the cleavage patterns of mRNAs were slightly different. As the tRNA insertion caused by gene rearrangement, the mature bicistronic mRNA *ND6/CYTB* found in *Bemisia tabaci* and *Maruca vitrata*, and the tricistronic mRNA *ATP8/ATP6/COIII* found in *M. vitrata* [47,48] were cleaved to *ND6*, *CYTB*, *ATP8/ATP6* and *COIII* in *S. houjii*. The monocistronic mRNAs *COI* and *ND3* were matured as the bicistronic mRNA *COI/ND3* in *S. houjii* since there was no tRNA insertion between them after gene rearrangement. Post-transcriptional cleavage processes of PCGs were affected by the tRNA inversion occurred downstream of the PCGs and insertion or removal between PCGs, rather than rearrangement of PCGs.

Sericothripinae includes three valid genera, *Sericothrips*, *Hydatothrips* and *Neohydotothrips*, of which a synapomorphic characteristic is the microtrichia rows dense in the abdominal tergite and sternite. In the phylogenetic tree, Sericothripinae is related to *Scricothrips* and *Echinothrips* [19,22], of which the microtrichia are also rows dense on abdominal tergites and sternites [49,50]. Based on this character, Sericothripini was recognized as a tribe together with Sericohripina and Scirtothripina (now placed in the *Scricothrips* genus-group) [49,51,52]. However, this character is also identified in other genra of Thripinae, such as the *Thrips*, *Mycterothrips* and *Projectothrips*, which are unrelated to *Sericothrips*, *Hydatothrips* and *Neohydotothrips* [19,21,23,52]. Microtrichia rows dense in abdominal tergite and sternite may not be the key characteristic for the tribe of Sericothripini or the subfamily of Sericothripinae. Moreover, limited mitogenomic resources and small sampling has hampered widely comparative studies for the phylogenetic status and taxonomic decision within Thripidae.

## 5. Conclusions

In summary, an insect mitogenome with extensive gene rearrangement is different in the transcriptional model, relative gene expression level and post-transcriptional cleavage process. The transcription and cleavage of the polycistronic transcript were not influenced when the location was interchanged among tRNAs or PCG, but the replicates of CR sequences will influence the transcription of the mitogenome, and the tRNA insertion or removal between PCGs and NCR in the 3′ end of PCG will influence the cleavage of polycistronic transcript. Our findings provide new insights into mitochondrial gene transcription, RNA processing and RNA degradation in insect mitogenomes with gene rearrangements and CR duplication.

## Figures and Tables

**Figure 1 insects-15-00700-f001:**
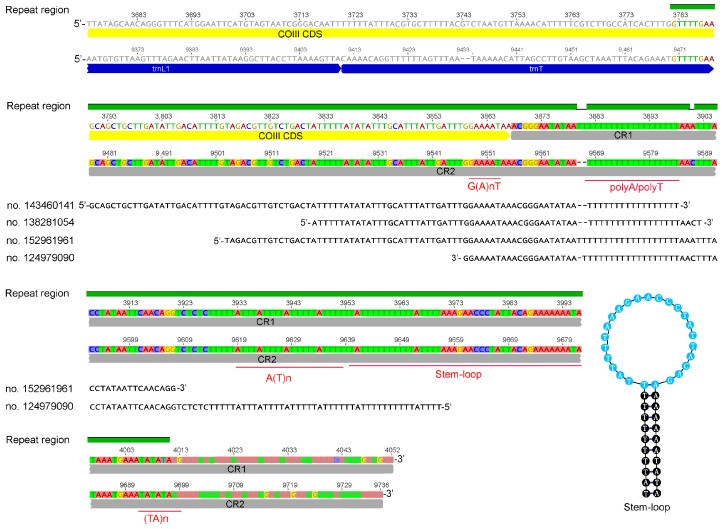
Comparative analysis of the control regions (CRs) and the ncRNAs transcribed from CRs of *Sericothrips houjii*. Green bold line indicates the repeat region, and nucleotides are colored. CRs are highlighted by colorful background and their transcripts are mapped below.

**Figure 2 insects-15-00700-f002:**
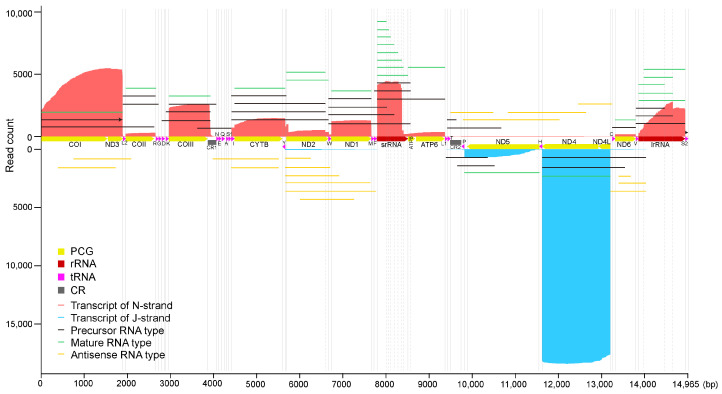
The quantitative transcription map of the *Sericothrips houjii* mitogenome. The mitogenome was arranged with the *x*-axis indicating J-strand orientation. Alignments of transcripts are indicated along the *y*-axis, with the transcripts of the N-strand and J-strand highlight by red and blue, respectively. The lines in green, black and yellow represent the mature, polycistronic and antisense types of transcripts, respectively. The lines with arrows indicate the same transcript.

**Figure 3 insects-15-00700-f003:**
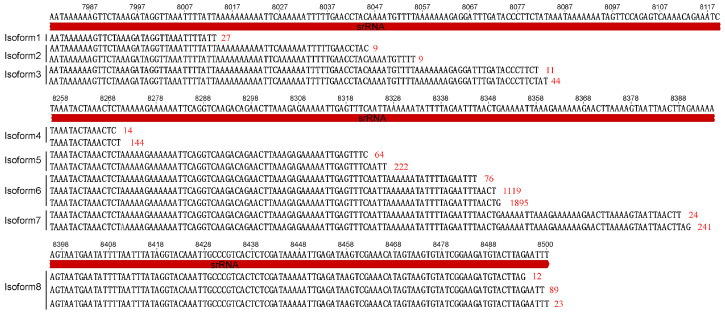
The isoforms of *srRNA* in the *Sericothrips houjii* mitogenome. The numbers of the transcripts with identical initiation and termination sites are highlighted in red.

**Figure 4 insects-15-00700-f004:**
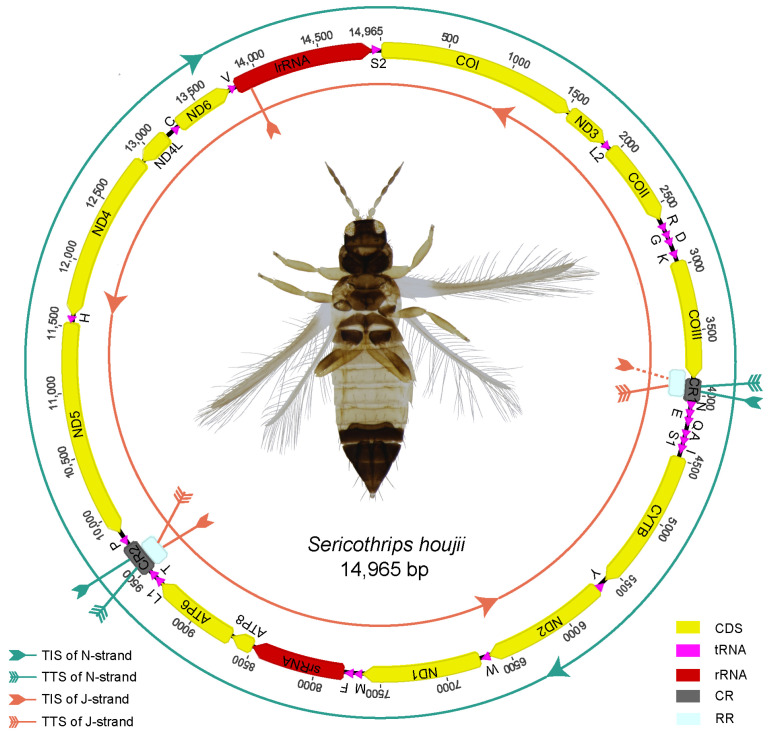
The proposed transcriptional model of the *Sericothrips houjii* mitogenomes. The green and orange line with arrow indicate the orientation of transcription of N-strand and J-strand, respectively.

**Figure 5 insects-15-00700-f005:**
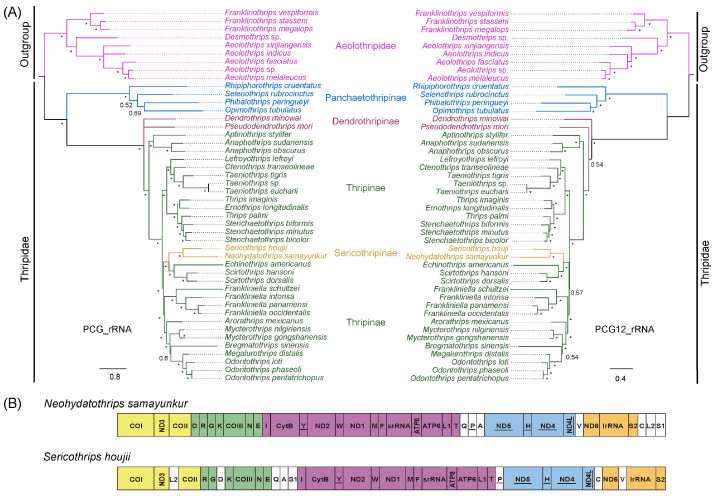
(**A**) Phylogenetic analysis based on the datasets PCG_rRNA and PCG12_rRNA. Different subfamilies are highlighted by colors. “*” indicates that the posterior probability is >90%. (**B**) Mitochondrial gene arrangement of *Neohydatothrips samayunkur* and *Sericothrips houjii*. The conserved gene blocks are highlighted by colors.

**Table 1 insects-15-00700-t001:** Comparison of the annotation of the *Sericothrips houjii* mitogenome based on DNA sequences and full-length transcripts.

Gene	Coding Strand	Annotation Using DNA Sequencing				Annotations According to Transcripts			
		Position	Length (bp)	Start Codon	Stop Codon	Position	Length (bp)	Strat Codon	Stop Codon
** *COI* **	J	1–1539	1539	AUA	UAA	1–1539	1539	AUA	UAA
** *ND3* **	J	1552–**1892**	**341**	AUU	**UA**	1552–**1891**	**340**	AUU	**U**
*trnL2*	J	1892–1955	64			1892–1955	64		
*COII*	J	1956–2618	663	AUU	UAA	1956–2618	663	AUU	UAA
** *trnR* **	J	**2665**–2718	**54**			**2664**–2718	**55**		
*trnG*	J	2719–2779	61			2719–2779	61		
*trnD*	J	2798–2858	61			2798–2858	61		
*trnK*	J	2897–2957	61			2897–2957	61		
** *COIII* **	J	**3070**–3867	**798**	AUA	UAA	**2959**–3867	**909**	AUA	UAA
CR1		3868–4052	185			3868–4052	185		
*trnN*	J	4053–4117	65			4053–4117	65		
*trnE*	J	4115–4179	65			4115–4179	65		
*trnQ*	J	4183–4251	69			4183–4251	69		
*trnA*	J	4291–4353	63			4291–4353	63		
*trnS1*	J	4354–4407	54			4354–4407	54		
*trnI*	J	4409–4477	69			4409–4477	69		
** *CYTB* **	J	**4482**–5588	**1107**	AUA	UAA	**4479**–5588	**1110**	AUA	UAA
*trnY*	N	5650–5586	65			5650–5586	65		
** *ND2* **	J	5678–**6656**	**979**	AUA	U	5678–**6659**	**982**	AUA	U
*trnW*	J	6660–6722	63			6660–6722	63		
** *ND1* **	J	**6729**–7641	**913**	**AUU**	U	**6723**–7641	**919**	**AUA**	U
*trnM*	J	7642–7702	61			7642–7702	61		
*trnF*	J	7712–7777	66			7712–7777	66		
** *srRNA* **	J	**7771**–**8504**	**734**			**7778**–**8500**	**723**		
*ATP8*	N	8501–8669	169	AUU	U	8501–8669	169	AUU	U
** *ATP6* **	J	**8636**–9346	**711**	**AUA**	UAA	**8696**–9346	**651**	**AUU**	UAA
*trnL2*	J	9347–9410	64			9347–9410	64		
** *trnT* **	J	**9418–9483**	**66**			**9411–9477**	**66**		
**CR2**		**9484**–9736	**253**			**9478**–9736	**259**		
*trnP*	N	9801–9737	65			9801–9737	65		
*ND5*	N	11,532–9856	1677	AUA	UAG	11,532–9856	1677	AUA	UAG
*trnH*	N	11,592–11,533	60			11,592–11,533	60		
*ND4*	N	12,896–11,593	1304	AUU	UAA	12,896–11,593	1304	AUU	UAA
*ND4L*	N	13,168–12,890	279	UAG	UAG	13,168–12,890	279	UAG	UAG
*trnC*	J	13,207–13,267	61			13,207–13,267	61		
** *ND6* **	J	**13,298**–13,762	**465**	**AUU**	UAA	**13,283**–137,62	**480**	**AUA**	UAA
*trnV*	J	13,764–13,819	56			13,764–13,819	56		
** *lrRNA* **	J	**13,819**–**14,907**	**1087**			**13,820**–**14,902**	**1083**		
*trnS1*	J	14,903–14,965	63			14,903–14,965	63		

Note: The modification of the annotation was bold to highlight.

## Data Availability

The annotated mitogenome sequence *Sericothrips houjii* has been deposited in the GenBank under the accession number PP697967. The full-length mitochondrial transcriptome of *Sericothrips houjii* has been deposited in the GenBank under the accession number PRJNA1130360.

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
