# Peer review of "Full-Length Transcriptome Profiling of the Complete Mitochondrial Genome of Sericothrips houjii (Thysanoptera: Thripidae: Sericothripinae) Featuring Extensive Gene Rearrangement and Duplicated Control Regions"

_insects, 2024, doi:10.3390/insects15090700_

Round 1

Reviewer 1 Report

Comments and Suggestions for Authors

The present manuscript reports the complete mitochondrial genome of Sericothrips houji by short-reads sequencing (Illumina Hiseq 6000 platform) and profiled by full-length transcriptome by long-reads sequencing. This is interesting and the first time to apply acquisition of mitochondrial transcriptome to correct gene boundaries in Thysanoptera. The genus Sericothrips is the type genus of the subfamily Sericothripinae in Thripidae. It is important that the authors selected the species Sericothrips houji in this study because there is only one species Neohydatothrips samayunkur( Sericothripinae) open in NCBI GenBank. These important results should be published. However, I do have the following comments in hope to improve the quality of the manuscript.

Comments to Authors:

 1.      It is suggested that “Thripidae” should be added in the title because fewer researchers knew Sericothripinae is the subfamily of Thripidae in the current classification.

 2.      There are some English mistakes in the text like line 100, 101,103 etc. Please check it carefully.

 3.      It is strongly suggested that the authors do phylogenetic analyses of Thysanoptera including Sericothrips houji. As mentioned, the genus Sericothrips is the type genus of the subfamily Sericothripinae in Thripidae and there is only one species of Sericothripinae open in NCBI GenBank.   The result will be helpful to learn the phylogenetic status of Sericothripinae in “Thripidae” and further consideration for taxonomic decision, like dividing Thripidae into several families, or redefining the subfamilies.  

 4. The gene order rearrangement in Figure 4, Table1, Figure2, Figure S1 are not consistent. Fig.4 is “D-R-G-K” (identical with those in Neohydatothrips samayunkur and Scritothrips dorsalis) but Table1, Figure2, Figure S1 is “RGDK”. Which one is correct?

 5.  Add the Schematic diagram of mitochondrial gene rearrangement of the close species for comparison after obtaining the phylogenetic results.

Comments on the Quality of English Language

There are some English mistakes in the text like line 100, 101,103 etc. Please check it carefully.

Author Response

Point 1: it is suggested that “Thripidae” should be added in the title because fewer researchers knew Sericothripinae is the subfamily of Thripidae in the current classification

Response: We have added “Thripidae” in the title.

Point 2: There are some English mistakes in the text like line 100, 101,103 etc. Please check it carefully.

Response: We have checked and corrected the similar grammatical problems of whole article.

Point 3: It is strongly suggested that the authors do phylogenetic analyses of Thysanoptera including Sericothrips houji. As mentioned, the genus Sericothrips is the type genus of the subfamily Sericothripinae in Thripidae and there is only one species of Sericothripinae open in NCBI GenBank. The result will be helpful to learn the phylogenetic status of Sericothripinae in “Thripidae” and further consideration for taxonomic decision, like dividing Thripidae into several families, or redefining the subfamilies

Response: We have added the phylogenetic analyses of Thripidae including Sericothripis houji. The corresponding abstract, introduction, methods, results and discussion were added in Line 47-48, Line 111-123, Line 189-205, Line 361-366 and Line 471-482, respectively. Corresponding, Figure 5A was exhibit the phylogenetic trees.

Point 4: the gene order rearrangement in Figure 4, Table1, Figure2, Figure S1 are not consistent. Fig.4 is “D-R-G-K” (identical with those in Neohydatothrips samayunkur and Scritothrips dorsalis) but Table1, Figure2, Figure S1 is “RGDK”. Which one is correct?

Response: The gene order trnR-trnG-trnD-trnK is correct. We have corrected it in Figure 4, and checked all annotations about this mitogenome to ensure the accurately.

Point 5: Add the Schematic diagram of mitochondrial gene rearrangement of the close species for comparison after obtaining the phylogenetic results.

Response: We added the schematic diagram in Figure 5B to comparison the mitochondrial gene arrangement between Sericothrips houji and its close species Neohydatothrips samayunkur. The comparison results are explained from Line 367 to Line 375.

Reviewer 2 Report

Comments and Suggestions for Authors

The topic of your study, investigating the full-length transcriptome profiling of the mitochondrial genome of Sericothrips houji, is novel and interesting. I would suggest editor of Insects accept this manuscript.

However, several points need clarification or improvement to strengthen the manuscript. The detailed comments were shown in the attached file.

Comments on the Quality of English Language

The English quality is good, Only a few minor errors were pointed out in the attached file. 

Author Response

Point 1: Line 17, “the transcription of the gene extensive rearranged mitogenomes“, the word “the gene” should be deleted

Response: We have deleted the “the gene” of this sentence.

Point 2: Line 27, Use “gene rearrangement“, or “rearranged genes” maybe the former is better. There might be other similar mistakes, please check.

Response: We have replaced the “extensive gene rearranged” with “extensive gene rearrangement” in line 18, line 27, line 100 and line 409.

Point 3: Line 122, “Berry Genomics”

Response: We have corrected “Berry Genomic” to “Berry Genomics”.

Point 4:       Line 125, Why are reads with '> 75 bp bases and a quality score < 3' removed? I mean, why use the parameter '> 75 bp'? Shouldn't all reads with a quality score < 3 be removed?

Response: Because the reads < 75 bp have been filtered. We have revised this sentence to “poly-Ns > 15 bp, < 75 bp in length and quality score < 3”.

Point 5: Table 1, The title of table 1 should be “Comparison of the annotation of…”. Please revise Table 1 to ensure the width and number of lines are standardized. Specifically, make the second line thinner in comparison to the other lines. Additionally, consider adding a thin separator line between the first and second lines to improve the clarity and readability of the table. The modified the annotations in table 1 could be highlighted in some way

Response: We modified the format of Table 1 followed the “insects-template” download from the official website of Insects, and added a thin to separate the first and second line. The modification of the annotations was bold to highlight.

Round 2

Reviewer 1 Report

Comments and Suggestions for Authors

The authors did efforts to revise their manuscript and did well in 

a short period. The revised MS  will reveal more information and contribute

it for the mitochondrial genomics , molecular phylogeny and taxonomy in thrips.  Well done! I would like to suggest the editor to accept this MS after correcting English errors.

Comments on the Quality of English Language

I am not an English native speaker, but I still found some obvious English errors (spelling and grammar). Please check it again.

For examples

line 101. Firstly, we precisely annotated......

line 105  molecule, real-time (SMRT)

line477  other genus should be other genera